

# The relationship between renal function indicators and preeclampsia in the second trimester of pregnancy: a retrospective study

Mingwei Li[1,*], Wei Liu[2,*], Xizhenzi Fan[3], Wenhui Song[3], Achou Su[3], Xue Zhang[4], Thomas Zheng[3] and Tianxiao Yu[3]

[1] Medical Affairs Department, the Fourth Hospital of Shijiazhuang, Shijiazhuang, Hebei Province, China
[2] Department of Breast Surgery, the Fourth Hospital of Shijiazhuang, Shijiazhuang, Hebei Province, China
[3] Research Center for Clinical Medical Sciences, the Fourth Hospital of Shijiazhuang, Shijiazhuang, Hebei Province, China
[4] Department of Public Health, the Fourth Hospital of Shijiazhuang, Shijiazhuang, Hebei Province, China
* These authors contributed equally to this work.

Corresponding author
Tianxiao Yu,
yutianxiao1111@163.com

## ABSTRACT

**Background:** To investigate the relationship between serum renal function indicators and preeclampsia (PE) in pregnant women at second trimester of pregnancy, determine the optimal critical values of the above indicators, and further identify the independent risk factors of PE.

**Methods:** We assessed the renal function indicators in second trimesters of 137 pregnant women with PE and 137 normal pregnant women who delivered at Shijiazhuang Fourth Hospital between January 2020 to December 2022. Paired *t*-tests, paired Mann-Whitney U tests, and Chi-square tests were used to evaluate differences of clinical data between the two groups. Receiver operating characteristic (ROC) curves was employed to establish the optimal critical values of the above indicators. A 1:1 matched case-control logistic regression analysis was conducted to identify the independent risk factors for PE.

**Results:** The levels of serum uric acid and the ratio of serum uric acid to serum creatinine were significantly higher in the PE group compared to the control ($P < 0.001$), while the serum creatinine levels were higher in control group ($P = 0.002$). The incidence of adverse maternal ($P < 0.001$) and neonatal outcomes ($P < 0.001$) in the PE group were higher than those in the control group. A ROC analysis based on the occurrence of PE showed that the levels of serum uric acid ($P < 0.001$), serum creatinine ($P = 0.006$) and the ratio of serum uric acid to serum creatinine ($P < 0.001$) were statistically significant. After adjusting for confounding factors, elevated serum uric acid (a $OR = 1.012$, 95% CI [1.005–1.019], $P < 0.001$) and an increased serum uric acid to serum creatinine ratio (a $OR = 1.190$, 95% CI [1.053–1.346], $P = 0.005$) were identified as independent risk factors for PE. There was no significant difference in renal function between maternal and newborn group in relation to the occurrence of adverse outcomes ($P > 0.05$ *vs.* all groups).

**Conclusions:** Through the analysis of renal function indicators in patients with PE in the second trimester of pregnancy and those in a normal control group, it is found

that elevated serum uric acid and serum uric acid to serum creatinine ratio in PE individual may serve as indicative markers for the onset of PE. Targeting this subset of the population for monitoring and management during the second trimester could enhance the efficacy of medical interventions.

# INTRODUCTION

Preeclampsia (PE) as the second most prevalent obstetrical complication, resulted in substantial maternal and neonatal morbidity and mortality (*Abalos et al., 2013*; *Ananth, Keyes & Wapner, 2013*; *Magee, Nicolaides & von Dadelszen, 2022*; *Myatt, 2022*; *Say et al., 2014*). PE usually occurs after 20 weeks of pregnancy. If not treated in time, it may cause serious or even fatal complications to pregnant women and fetuses (*Davidson et al., 2021*). Therefore, early prediction and prevention of PE is very important and necessary. Predicting the incidence of PE during the first or second trimester of pregnancy has been a research hotspot for decades. However, the ideal predictive markers with high sensitivity and specificity are not clear. PE prediction using clinical risk factors such as obesity and hyperglycemia has been widely accepted and practiced, but medical history-based prediction only identifies approximately 30% patients who will eventually develop PE (*Bartsch et al., 2016*; *Leslie, Thilaganathan & Papageorghiou, 2011*).

Although novel screening modalities including angiogenic modulators and uterine artery Doppler velocimetry are gaining momentum in some developed regions, their low positive predicative value, standardization and expenses are concerning, especially in resource poor areas, their implementation will be difficult.

One of the characteristics of PE is vascular endothelial cell dysfunction, which can lead to reduced blood flow and subsequently result in kidney damage. Additionally, patients with PE exhibit elevated levels of inflammatory cytokines and increased oxidative stress, which may indirectly contribute to kidney injury. The renal function indicatiors were readily accessible. Several studies have shown that there is a correlation between renal function indicators and PE in pregnant patients (*Bellos & Pergialiotis, 2020*; *Hawkins et al., 2012*). However, a systematic review has also shown that some renal function indicators, such as uric acid, is not clinically useful due to its low predicative value (*Cnossen et al., 2006*). Our objective is to investigate whether the concentration of pregnant woman renal function indicators in the second trimester of pregnancy were correlated with the occurrence of PE in late pregnancy, then identify the independent risk factors of PE.

# MATERIALS AND METHODS

This is a single-center, retrospective, case-control study. The subjects of the study were pregnant patients who were diagnosed with PE and received prenatal care at the Fourth Hospital of Shijiazhuang, China, from January 2020 to December 2022. The inclusion criteria included singleton pregnant patients, aged 18-45 years who underwent renal

function tests between the gestational age of 14–27$^{+6}$ weeks. The exclusion criteria included the following: (1) multiple pregnancy; (2) second trimester miscarriage; (3) pre-existing autoimmune disease, chronic hypertension, thyroid dysfunction, or diabetes before pregnancy; (4) abnormal renal function before pregnancy; (5) hypertension or proteinuria occurred before the renal function test in the second trimester of pregnancy. Gather test results from the electronic medical records system.

All pregnant women and their families were thoroughly informed about the purpose of the study and provided written informed consent at the time of enrollment. The study was conducted in accordance with the principles of Declaration of Helsinki and received approval from the Ethics Committee of the Fourth Hospital of Shijiazhuang (20230149).

### Definition

PE was defined as having a blood pressure of 140/90 mmHg or higher on two occasions at least 4 h after 20 weeks of gestation, or a blood pressure of 160/100 mmHg or higher, with proteinuria a of ≥300 mg/24 h (or at least 2+ on urine dipstick test). In the absence of proteinuria, PE can be diagnosed as new-onset hypertension along with any of the following conditions: thrombocytopenia, impaired liver function, renal insufficiency, pulmonary edema, new-onset headache (*American College of Obstetricians and Gynecologists, 2020*).

### Statistical analysis

Statistical evaluations were performed with SPSS and MedCalc, while graphical representations were created utilizing GraphPad and R software. The control group was selected with a 1:1 ratio, matched to the case group based on age and gestational at the second trimester examination. Paired *t*-tests were conducted for continuous variables, while chi-square tests were utilized for categorical variables. To evaluate the predictive performance, we employed ROC curve analysis. Using a 1:1 matched case-control study design, logistic regression analysis was conducted to determine the association of the indicators with PE. A *P*-value of less than 0.05 was considered statistically significant.

## RESULTS

### Comparison of renal function indicators between PE group and control group in the second trimester of pregnancy

Ultimately, our study included 137 pregnant women diagnosed with PE and 137 age-matched controls (*P* = 0.374) who were also matched for gestational age at the time of the mid-pregnancy examination (*P* = 0.696) (Table 1). The parity between the two groups (*P* = 0.306) did not reveal any statistically significant differences. The PE group had a significantly higher proportion of patients with gestational hypertension history (7.3% *vs.* 0.0%, *P* < 0.001) and a family history of hypertension (48.2% *vs.* 21.1%, *P* < 0.001) compared to the control group. The pre-pregnancy body mass index (BMI) in PE group was higher than that of the control group (median: 23.23 kg/m$^2$ *vs.* 21.77 kg/m$^2$, *P* < 0.001). There were no significant differences in BUN levels (median: 2.60 mmol/L *vs.* 2.80 mmol/L, *P* = 0.151) between the two groups. However, compared to control group,

**Table 1 General information of participants.**

|  | Preeclampsia ($n$ = 137) | Control ($n$ = 137) | $P$-value |
|---|---|---|---|
| Age (years) | 30.31 (27.79, 33.41) | 30.52 (27.65, 33.06) | 0.374 |
| Primigravida | 92 (67.2) | 87 (63.5) | 0.306 |
| Pre-pregnancy BMI, kg/m$^2$ | 23.23 (20.64, 26.77) | 21.77 (19.41, 23.88) | <0.001 |
| History of gestation hypertensive | 10 (7.3) | 0 (0.0) | 0.001 |
| History of family hypertension | 66 (48.2) | 29 (21.1) | <0.001 |
| Gestational age at second trimester test | 26.00 (25.43, 26.71) | 26.00 (25.43, 26.79) | 0.696 |

Note:
BMI, body mass index. History of gestation hypertensive including pre-eclampsia and gestational hypertension. Data that conformed to a normal distribution were expressed as mean ± standard deviation and analyzed using paired t-tests. Non-normally distributed data were reported as median (lower quartile, upper quartile), and compared using paired rank sum tests. Difference between categorical variables was assessed using chi-square test.

the PE group exhibited lower levels of serum creatinine (median: 44.90 μmol/L $vs.$ 46.80 μmol/L, $P$ = 0.002), higher serum uric acid levels (mean: 226.95 μmol/L $vs.$ 197.85 μmol/L, $P$ < 0.001), and a higher serum uric acid to creatinine ratio (SUA/sCr) (mean:4.85 $vs.$ 4.15, $P$ < 0.001), with these differences being statistically significant (Fig. 1).

## Comparison of adverse pregnancy outcomes between PE and control groups

The gestational age at delivery was lower in the PE group compared to the control group (median: 36.43 weeks $vs.$ 39.29 weeks, $P$ < 0.001). In the PE group, 16.8% of the patients were diagnosed with early-onset PE. There was no difference in the newborn gender between the two groups ($P$ = 0.167). The birth weight (mean: 2,443.98 g $vs.$ 3,284.42 g, $P$ < 0.001) and newborn length (median: 48.00 cm $vs.$ 50.00 cm, $P$ < 0.001) in the PE group were lower than those in the control group. Most patients in the PE group delivered by cesarean section (87.6% $vs.$ 40.9%, $P$ < 0.001). The rates of adverse maternal (17.5% $vs.$ 1.5%, $P$ < 0.001) and neonatal outcomes (62.0% $vs.$ 10.2%, $P$ < 0.001) were higher in the PE group compared to the control group (Table 2).

## The predictive value of renal function indicators for PE

In order to evaluate the predictive utility of renal functional indices of the development of PE during the mid-trimester of gestation. We carried out a ROC curve analysis. The findings indicated that serum creatinine and uric acid levels of pregnant patients during the second trimester have the potential to differentiate between PE and the control group. In this analysis, the area under the curve (AUC) for uric acid was 0.635, with a 95% confidence interval (CI) of [0.575 to 0.692] ($P$ < 0.001). The AUC for serum creatinine was 0.595, with a 95% CI of [0.534 to 0.653] ($P$ = 0.006). The AUC for BUN was 0.550, with a 95% CI of [0.489–0.612] ($P$ = 0.157). Meanwhile, the AUC for SUA/sCr was 0.677, with a 95% CI of [0.618–0.732] ($P$ < 0.157). These findings suggested that uric acid and serum creatinine may be a clinical value of PE (Fig. 2).

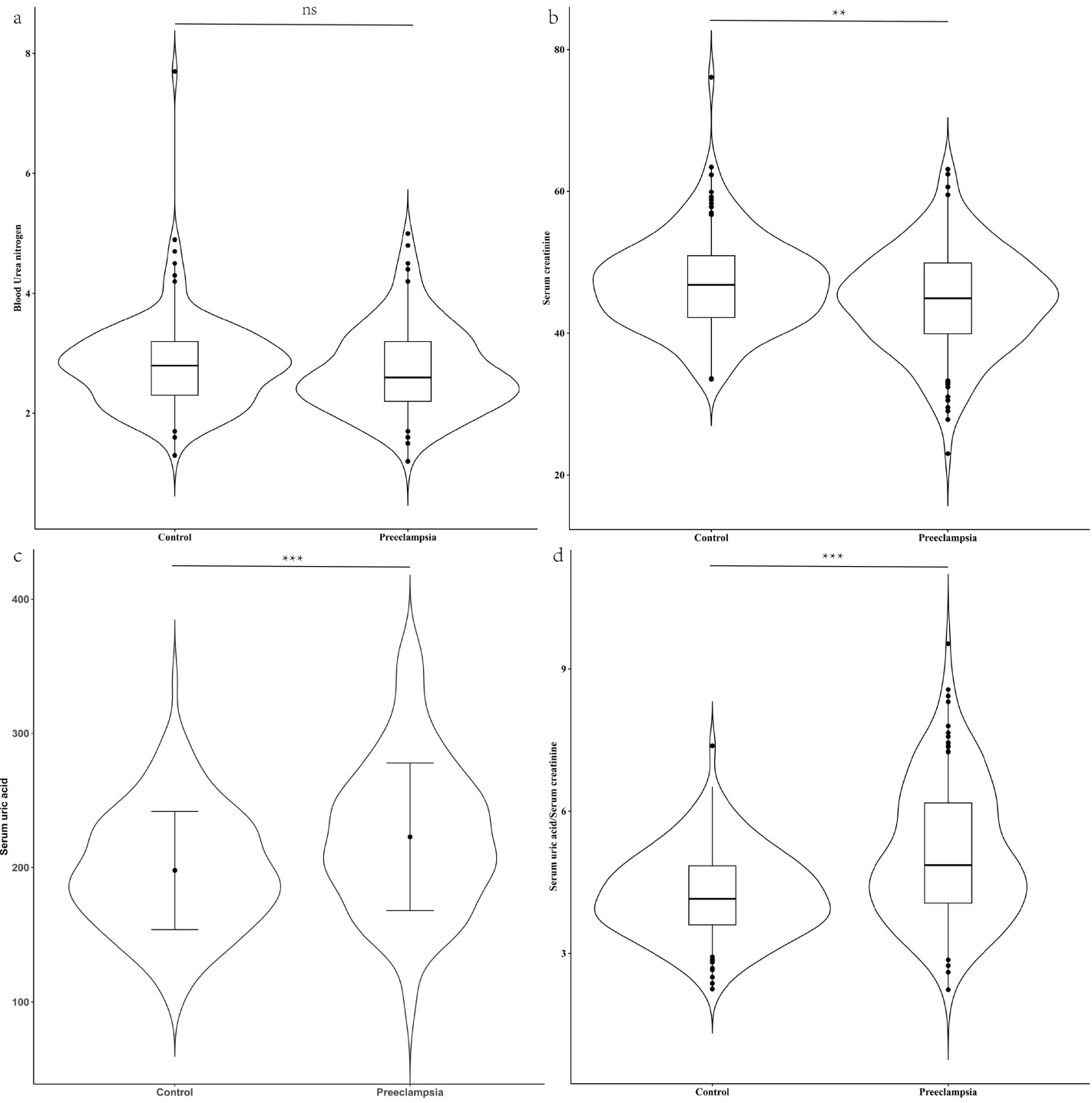

**Figure 1   Renal function between case group and control group in the second trimester.** (A) Distribution of blood urea nitrogen; (B) distribution of serum creatinine; (C) distribution of serum uric acid; (D) distribution of serum uric acid/Serum creatinine. Blood urea nitrogen, serum creatinine and the ratio of serum uric acid to serum creatinine, which were non-normally distributed, are presented as violin plot. The scatter points indicate values below the 5th percentile and above the 95th percentile. The difference between the two groups was evaluated using a paired rank sum test. Conversely, serum uric acid had a normal distribution and is presented as mean ± standard deviation. The difference between the two groups was assessed with a paired t-test. ns indicate no statistical difference; **$P < 0.01$ indicate a statistical difference; ***$P < 0.001$ indicate a statistical difference.

**Table 2 Characteristics of maternal and neonatal birth outcomes.**

|  | Preeclampsia (*n* = 137) | Control (*n* = 137) | *P*-value |
|---|---|---|---|
| Gestational age, weeks | 36.43 (34.57, 37.71) | 39.29 (38.50, 39.93) | <0.001 |
| Premature birth (<34 weeks) | 23 (16.8) | 1 (0.7) | <0.001 |
| Premature birth (<37 weeks) | 80 (58.4) | 6 (4.4) | <0.001 |
| Newborn gender, Male | 65 (47.4) | 74 (54.0) | 0.167 |
| Birth weight, g | 2,443.98 ± 689.12 | 3,284.42 ± 448.53 | <0.001 |
| Newborn length | 48.00 (45.00, 49.00) | 50.00 (50.00, 51.00) | <0.001 |
| Delivery mode |  |  |  |
| Vaginal delivery | 17 (12.4) | 81 (59.1) | <0.001 |
| Cesarean delivery | 120 (87.6) | 56 (40.9) |  |
| Adverse maternal outcomes | 24 (17.5) | 2 (1.5) | <0.001 |
| Placental abruption | 11 (8.0) | 1 (0.7) | 0.005 |
| HELLP syndrome | 11 (8.0) | 0 (0.0) | <0.001 |
| Retinal detachment | 6 (4.4) | 0 (0.0) | 0.030 |
| Postpartum hemorrhage | 2 (1.5) | 1 (0.7) | >0.999 |
| Adverse neonatal outcomes | 85 (62.0) | 14 (10.2) | <0.001 |
| Respiratory distress syndrome | 55 (40.1) | 2 (1.5) | <0.001 |
| Small for gestational age | 52 (38.0) | 10 (7.3) | <0.001 |
| Intracranial hemorrhage | 18 (13.1) | 2 (1.5) | <0.001 |
| Neonatal infection | 24 (17.5) | 0 (0.0) | <0.001 |
| Neonatal asphyxia | 5 (3.6) | 0 (0.0) | 0.030 |

Notes:
HELLP syndrome, hemolysis, elevated liver enzymes, and low platelet syndrome. Data that conformed to a normal distribution were expressed as mean ± standard deviation and analyzed using paired *t*-tests. Non-normally distributed data were reported as median (lower quartile, upper quartile), and compared using paired rank sum tests. Difference between categorical variables was assessed using chi-square test.
Adverse maternal outcomes including placental abruption, HELLP syndrome, retinal detachment or postpartum hemorrhage; Adverse neonatal outcomes including respiratory distress syndrome, small for gestational age, intracranial hemorrhage, neonatal infection or neonatal asphyxia.

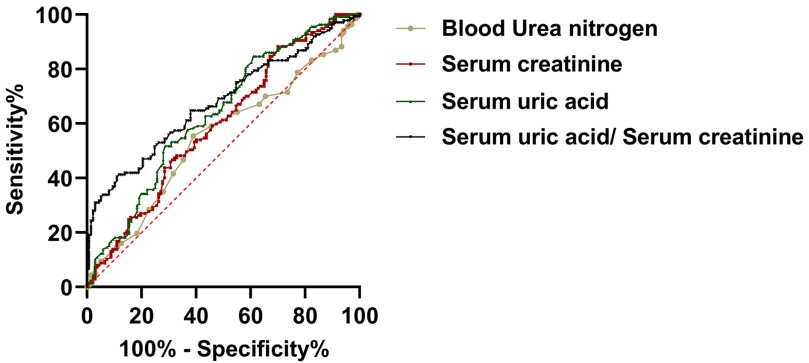

**Figure 2 The predictive value of renal function indicators for preeclampsia.**

## Evaluation of renal functions as predictive indicators

We performed a case-control matching logistic regression analysis on BUN, serum creatinine, and serum uric acid, taking into account factors such as a history of gestational

**Table 3 Evaluation of renal functions as predictive indicators.**

|  | OR (95% CI) | P-value | aOR (95% CI) | P-value |
|---|---|---|---|---|
| Blood Urea nitrogen, mmol/L | 0.797 [0.563–1.128] | 0.201 | 0.651 [0.419–1.011] | 0.056 |
| Serum creatinine, μmol/L | 0.943 [0.906–0.980] | 0.003 | 0.960 [0.919–1.002] | 0.064 |
| Serum uric acid, μmol/L | 1.011 [1.005–1.016] | <0.001 | 1.012 [1.005–1.019] | <0.001 |
| Serum uric acid/ Serum creatinine | 1.266 [1.128–1.422] | <0.001 | 1.190 [1.053–1.346] | 0.005 |

Note:
aOR, adjust odds ratio; History of gestational hypertension, History of family hypertension, pre-pregnancy body mass index. By 1:1 matched case-control matching logistic regression analysis.

hypertension, a family history of hypertension, and pre-pregnancy BMI (Table 3). There was no significant statistical difference between BUN ($aOR = 0.651$, 95% CI [0.419–1.011], $P = 0.056$) and serum creatinine ($aOR = 0.960$, 95% CI [0.919–1.002], $P = 0.064$) in the two groups. Serum uric acid levels during the second trimester of pregnancy ($aOR = 1.012$, 95% CI [1.005–1.019], $P < 0.001$) were determined as an independent factor for the development of PE. Additionally, the ratio of serum uric acid to serum creatinine in the second trimester ($aOR = 1.190$, 95% CI [1.053–1.346], $P = 0.005$) was identified as an independent predictive factor against PE.

Serum uric acid levels were categorized into a high group (>238 μmol/L) and a low group (≤238 μmol/L), based on the optimal cut-off determined by the ROC. A case-control matching logistic regression analysis revealed an odds ratio of 1.717 (95% CI [1.217–2.424], $P < 0.001$). SUA/sCr were categorized into a high group (>5.33) and a low group (≤5.33), based on the optimal cut-off determined by the ROC. A case-control matching logistic regression analysis revealed an odds ratio of 1.954 (95% CI [1.389–2.750], $P < 0.001$).

## Distribution of renal function indicators among different delivery outcomes

We divided PE patients into two groups according to adverse pregnancy outcomes. To ensure a balanced comparison, the groups were meticulously matched for age and gestational age at the time of the second trimester examination with the cohort that did not encounter adverse outcomes. There were 24 cases in the adverse maternal outcomes group and 45 cases in the adverse neonatal outcomes group (Table 4).

Pregnant women in the adverse outcome group also accompanied with newborn adverse outcome (78.3% vs. 26.1%, $P = 0.001$) and premature delivery (mean: 34.53 weeks vs. 37.47 weeks., $P = 0.001$). There were no significant differences in pre-pregnancy BMI (mean: 25.67 kg/m² vs. 24.50 kg/m², $P = 0.482$), BUN levels (mean: 2.91 mmol/L vs. 2.57 mmol/L, $P = 0.118$), serum creatinine levels (mean: 45.49 μmol/L vs. 44.65 μmol/L, $P = 0.727$), serum uric acid levels (mean: 221.96 μmol/L vs. 227.05 μmol/L, $P = 0.658$), and SUA/sCr (mean: 5.03 vs. 5.21, $P = 0.685$) in the two groups.

Features in the neonatal adverse outcome group were also associated with early delivery in terms of gestational weeks (median: 35.43 weeks vs. 37.71 weeks, $P < 0.001$). There were no significant differences in the adverse maternal outcomes (24.4% vs. 11.1%, $P = 0.167$), pre-pregnancy BMI (median: 23.44 kg/m² vs. 22.21 kg/m², $P = 0.135$), BUN (median: 2.60

**Table 4 Distribution of renal function indicators among cases with or without adverse outcomes.**

|  | With adverse maternal outcomes (*n* = 23) | Without adverse maternal outcomes (*n* = 23) | *P*-value |
|---|---|---|---|
| Age (years) | 29.11 (27.62, 33.77) | 28.98 (27.63, 33.14) | 0.553 |
| Gestational age at second trimester test | 26.23 ± 0.92 | 26.21 ± 0.92 | 0.773 |
| Gestational age, weeks | 34.53 ± 2.85 | 37.47 ± 2.35 | 0.001 |
| Pre-pregnancy BMI, kg/m$^2$ | 25.76 ± 5.47 | 24.50 ± 5.78 | 0.482 |
| Adverse neonatal outcomes | 18 (78.3) | 6 (26.1) | 0.001 |
| Blood urea nitrogen, mmol/L | 2.91 ± 0.80 | 2.57 ± 0.70 | 0.118 |
| Serum creatinine, μmol/L | 45.49 ± 6.86 | 44.65 ± 8.11 | 0.727 |
| Serum uric acid, μmol/L | 221.96 ± 47.22 | 227.05 ± 58.60 | 0.658 |
| Serum uric acid/Serum creatinine | 5.03 ± 1.53 | 5.21 ± 1.62 | 0.685 |
|  | **With adverse neonatal outcomes (*n* = 45)** | **Without adverse neonatal outcomes (*n* = 45)** | ***P*-value** |
| Age (years) | 29.92 ± 2.89 | 29.92 ± 2.89 | 0.924 |
| Gestational age at second trimester test | 25.98 ± 0.79 | 25.99 ± 0.81 | 0.810 |
| Gestational age, weeks | 35.43 (33.43, 36.57) | 37.71 (37.14, 39.57) | <0.001 |
| Pre-pregnancy BMI, kg/m$^2$ | 23.44 (20.69, 27.32) | 22.21 (20.17, 25.96) | 0.135 |
| Adverse maternal outcomes | 11 (24.4) | 5 (11.1) | 0.167 |
| Blood Urea nitrogen, mmol/L | 2.60 (2.20, 3.20) | 2.60 (2.20, 3.20) | 0.806 |
| Serum creatinine, μmol/L | 44.28 ± 6.75 | 45.44 ± 8.78 | 0.438 |
| Serum uric acid, μmol/L | 227.33 ± 61.37 | 223.69 ± 47.00 | 0.746 |
| Serum uric acid/Serum creatinine | 5.25 ± 1.63 | 5.08 ± 1.35 | 0.689 |

**Note:**

Data that conformed to a normal distribution were expressed as mean ± standard deviation and analyzed using paired *t*- tests. Non-normally distributed data were reported as median (lower quartile, upper quartile), and compared using paired rank sum tests. Difference between categorical variables was assessed using chi-square test. Adverse maternal outcomes including placental abruption, hemolysis, elevated liver enzymes, and low platelet syndrome, retinal detachment or postpartum hemorrhage; Adverse neonatal outcomes including respiratory distress syndrome, small for gestational age, intracranial hemorrhage, neonatal infection or neonatal asphyxia.

mmol/L *vs*. 2.60 mmol/L, *P* = 0.806), serum creatinine (mean: 44.28 μmol/L *vs*. 45.44 μmol/L, *P* = 0.438), serum uric acid (mean: 227.33 μmol/L *vs*. 223.69 μmol/L, *P* = 0.746) and SUA/sCr (mean: 5.25 *vs*. 5.08, *P* = 0.689).

# DISCUSSION

Compared to the first trimester, the fetus experiences accelerated growth during the second trimester, which is accompanied by a gradual metabolic transition of the mother. We believe that the level of sensitivity is higher during the second trimester compared to the first trimester.

Through the analysis of renal function indicators in pregnant women during their second trimester, we found that elevated uric acid and SUA/sCr may be related to the onset of PE in late pregnancy. However, for pregnant women with PE, there is no difference in BUN, serum creatinine, and serum uric acid levels during the second trimester, regardless of whether adverse maternal or neonatal outcomes occur.

Several studies, including those from the National Institute for Health and Care excellence, the American College of Obstetricians and Gynecologists, and Townsend, have utilized demographic information and disease history to identify common predictive indicators (*National Institute for Health and Care Excellence, 2022*; *Townsend et al., 2019*).

This method facilitates the identification of individuals at high risk for hypertension, thereby allowing for focused attention and care, through a meta-analysis, found that factors such as blood pressure, bilateral notching, and pulsatility index during the second trimester improves the identification of nulliparous women at risk for PE (*Kleinrouweler et al., 2013*). Other studies have also utilized indicators such as placental growth factors as markers for PE (*O'Gorman et al., 2017*). However, the application of these indicators may be may be limited by technology or financial resources, and may not be feasible in primary medical institutions. This study analyzed the renal function indicators of pregnant women of the same age and gestational age. We found that the relative increase of uric acid level in the second trimester of pregnancy was an independent risk factor for PE. Additionally, some studies also found that serum uric acid was a risk factor for severe PE in the third trimester (*Seow et al., 2005*; *Voto et al., 1988*).

Several studies have reported that PE can result in renal injury, potentially leading to the increase of uric acid levels. This is due to the fact that renal injury typically manifests as glomerular endotheliosis, resulting in a decrease glomerular filtration rate and renal blood flow, thereby reducing the uric acid clearance. Although clinical symptoms of preeclampsia typically appear after 20 weeks of gestation, maternal endothelial cell dysfunction precedes these clinical manifestations. Endothelial cell dysfunction can facilitate the adhesion and infiltration of inflammatory cells, enhance the inflammatory response in the kidney, and damage the glomerular filtration barrier, leading to impaired kidney function. Additionally, oxidative stress can damage vascular endothelial cells, resulting in abnormal vasoconstriction and reduced kidney blood flow, which in turn affects kidney function. Oxidative stress can also promote the proliferation of vascular smooth muscle cells, intensify vascular contraction, and further reduce kidney blood flow, thereby exacerbating kidney function impairment (*Assali et al., 1953*; *Chen et al., 2016*; *McCartney, Spargo & Lorincz, 1964*; *Moran et al., 2003*; *Piani et al., 2023*; *Sarles et al., 1968*; *Stratta et al., 1987*). Various reports have suggested that serum uric acid levels are highly elevated during the first trimester in pregnant women who subsequently developed PE. The elevation in serum uric acid levels observed in PE has been attributed to diminished uric acid excretion in the proximal tubules, a phenomenon secondary to hypovolemia. This may occur early in the development of PE (*Wu et al., 2012*). In a systematic review, *Cnossen et al. (2006)* reported that uric acid levels in the second trimester were not definitive in predicting PE due to their low sensitivity and specificity. Our results show the same phenomenon. Nevertheless, following adjustment in the logistic regression model, both serum uric acid levels and the ratio of serum uric acid to serum creatinine persisted as independent risk factors. We contend that in areas with less access to advanced medical care, serum uric acid levels and SUA/sCr ratio can serve as valuable early warning signs. *Chen et al. (2016)* concluded that increase in serum uric acid at the onset of PE could be a result of a maternal response, rather than a promoter of the disease. In our study, we excluded the pregnant women with other diseases. We found that even in the absence of hypertension, pregnant women who eventually developed PE had elevated uric acid levels compared to those who did not develop PE.

We carried out a paired analysis based on maternal age and gestational week at the second trimester check-up, minimizing the influence of pregnancy metabolism (*Lopes van Balen et al., 2019*) and maternal age (*Zhou et al., 2023*) on the outcomes. A significant strength of our study is that we utilized established markers from the second trimester to identify pregnant women at elevated risk of developing s-PE. We observed differences in renal indicators between the PE group and control group. Our study also has some limitations due to a relatively small patient sample and its retrospective nature. In addition, despite our efforts to control for confounding factors such as maternal age, gestational age detection, maternal history of hypertension (*Mustary et al., 2024*), family history of hypertension (*Kay, Wedel & Smith, 2021*), and pre-pregnancy BMI, it is possible that unmeasured variables, such as lifestyle habits and environmental factors, or unforeseen confounders could influence the results. Based on the presented results, serum uric acid can be considered as a potential risk indicator. The data from this study suggest an optimal cut-off value of 238 mmol/L for serum uric acid. However, it requires validation and determination through large-sample, multi-center cohort studies. Given the moderate AUC value, the utility of this indicator in isolation is quite restricted. It is recommended that future efforts focus on developing a composite model that incorporates additional biomarks to enhance the diagnostic efficacy.

## CONCLUSIONS

Through the analysis of second trimester assessments in asymptomatic PE patients, we identified and matched a cohort of patients with comparable age and gestational age. Furthermore, following logistic regression analysis and adjustment, we found that an increase in uric acid levels in the second trimester was a risk factor for PE. This implies that although PE potentially causing damage to various organs, indicators of renal function remain sensitive in the second trimester of pregnancy. Therefore, clinicians can regard them as risk factors in their decision-making process.

## ACKNOWLEDGEMENTS

We thank all the subjects and participants for their support, cooperation, and involvement in this study.

### Funding

This work was funded by the Key Medical Scientific Research Project of Hebei province (No. 20240716). The funders had no role in study design, data collection and analysis, decision to publish, or preparation of the manuscript.

### Grant Disclosures

The following grant information was disclosed by the authors:
Key Medical Scientific Research Project of Hebei province: 20240716.

## Competing Interests

The authors declare that they have no competing interests.

## Author Contributions

- Mingwei Li conceived and designed the experiments, performed the experiments, analyzed the data, prepared figures and/or tables, and approved the final draft.
- Wei Liu conceived and designed the experiments, performed the experiments, prepared figures and/or tables, and approved the final draft.
- Xizhenzi Fan performed the experiments, analyzed the data, prepared figures and/or tables, and approved the final draft.
- Wenhui Song performed the experiments, analyzed the data, prepared figures and/or tables, and approved the final draft.
- Achou Su performed the experiments, analyzed the data, prepared figures and/or tables, and approved the final draft.
- Xue Zhang conceived and designed the experiments, performed the experiments, prepared figures and/or tables, and approved the final draft.
- Thomas Zheng conceived and designed the experiments, performed the experiments, authored or reviewed drafts of the article, and approved the final draft.
- Tianxiao Yu conceived and designed the experiments, performed the experiments, authored or reviewed drafts of the article, and approved the final draft.

## Human Ethics

The following information was supplied relating to ethical approvals (*i.e.*, approving body and any reference numbers):

The ethics committee of The Fourth Hospital of Shijiazhuang (No. 20230149).

## Data Availability

The raw measurements are available in the Supplemental File.

## Supplemental Information

Supplemental information for this article can be found online at http://dx.doi.org/10.7717/peerj.19027#supplemental-information.

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
