# Peer review of "The relationship between renal function indicators and preeclampsia in the second trimester of pregnancy: a retrospective study"

_PeerJ, doi:10.7717/peerj.19027_

## Round 0.1 · original submission · Major Revisions

Dear authors,

Manuscript titled "The relationship between renal function indicators and preeclampsia in the second trimester of pregnancy" that you submitted to PeerJ has been reviewed.

The reviewer(s) have suggested that some important points must be clarified and have requested substantial changes to be made in the manuscript. Therefore, I invite you to respond to the reviewer(s)' comments and revise your manuscript. The reviewer(s) comments are included at the end of this letter.

Please ensure that all review, editorial, and staff comments are addressed in a response letter and that any edits or clarifications mentioned in the letter are also inserted into the revised manuscript where appropriate.

Reviewer 1 ·

Basic reporting

no comment

Experimental design

no comment

Validity of the findings

no comment

Additional comments

Summary

In this matched case-control study, which includes 137 PE cases and 137 controls, the relationship between renal function indicators—specifically, serum uric acid and serum creatinine—and preeclampsia (PE) is assessed. According to the results, high serum uric acid could be a possible PE biomarker. Its minimal visual contrast in data display and moderate predictive accuracy, however, cast doubt on its generalizability and usefulness.

Strengths

This study targets a critical area of maternal health, exploring cost-effective and widely available biomarkers for early prediction of PE, which could be impactful for clinical practice, particularly in low-resource settings.
The matched case-control design minimizes confounding and enhances the reliability of the findings.
Robust statistical approaches, including logistic regression and ROC analysis, strengthen the validity of the study results.
Highlighting low-cost biomarkers like serum uric acid makes the findings practical and relevant in resource-constrained environments.

Recommendations

The box plots in Figure 1 barely show any noticeable differences between cases and controls, undermining the visual impact of the data. Moreover, the box plots do not make a compelling argument for significant differences. Is the data overfitted or inherently less discriminative?
How can such small differences be considered meaningful for clinical applications? Are the findings robust to larger cohorts or external validation?
The ROC curve in Figure 2 demonstrates weak predictive performance for both serum uric acid (AUC = 0.635) and serum creatinine (AUC = 0.595). How can these biomarkers be positioned as useful predictive tools with such low AUC values? Would combining biomarkers into a composite model improve prediction accuracy?
The manuscript lacks a detailed explanation of the biological pathways linking renal function markers to preeclampsia.
Why not discuss the role of oxidative stress, endothelial dysfunction, or hyperuricemia in more detail?
Although serum uric acid was identified as an independent risk factor, the effect size is minimal (AOR = 1.012). Is this clinically meaningful?
Stratification by PE severity (early vs. late-onset) or by other patient demographics (e.g., BMI, parity) could reveal more nuanced patterns.
The relatively small, single-center sample limits the applicability of the results to broader populations. How do the authors plan to validate these findings in a multicenter or larger cohort study?
The manuscript does not outline a clear strategy for translating these findings into clinical practice. How would serum uric acid measurement be integrated into routine prenatal care? Are there practical recommendations or cutoff values for clinicians?
The manuscript contains several grammatical and stylistic issues, such as “serum uric acid levels were higher in control group.” A thorough language edit is strongly recommended.

Conclusion

An essential first step in investigating renal function markers as possible preeclampsia predictors is provided by this study. Nevertheless, its significance is lessened by the slight variations in marker levels, poor ROC performance, and scant biological description. By enhancing data visualization, investigating new biomarkers or composite models, and making a stronger argument for clinical significance, the authors should overcome these shortcomings. Important changes must be made before publishing.

Reviewer 2 ·

Basic reporting

Overall, the manuscript is well-structured and well-written. The authors clearly laid out their hypothesis in the introduction and included related results. The tables and figures are in good shape.

Experimental design

The study used a matched case-control design, aiming to remove confounders. However, it is unclear how the matching variables were selected and why only a few were matched. Please justify whether it is necessary to do propensity score matching. Based on a matched case-control design, the statistical analysis was appropriate to answer the research questions.

Moreover, although the p-value for AUC is significant for uric acid, the point estimation of AUC is still relatively low. I recommend adding more variables or using more powerful predictive models to increase the predictive ability for early diagnosis.

Validity of the findings

This work targets an important research question about indicators of preeclampsia (PE) in pregnant women. The underlying data is provided. Conclusions are well stated with support. However, as I stated before, the AUC is too low to be a useful prediction model.

·

Basic reporting

The article presents a well-structured investigation into the correlation between renal function indicators in the second trimester of pregnancy and the occurrence of pre-eclampsia (PE).

The language used is clear and professional. However, some issues should be solved: "The increased in serum uric acid levels"; "predicating the incidence of PE"; "MedClac"; "There parity"; "emerged as independent protective factors for PE" predictive instead of protective; "progressing to s-PE" (ambiguous phrasing); "However, after adjusting for logistic regression"; "in regions where medical facilities are relatively inadequate" (inadequate may sound not respectful); "History of gestation hypertensive including pre-eclampsia and"; "By analyzing the results of the second trimester examinations of asymptomatic PE patients, We selected patients"

Experimental design

Please consider incorporating recent publications, such as Piani, F., Agnoletti, D., Baracchi, A., Scarduelli, S., Verde, C., Tossetta, G., ... & Borghi, C. (2023). Serum uric acid to creatinine ratio and risk of preeclampsia and adverse pregnancy outcomes. Journal of Hypertension, 41(8), 1333-1338. Based on your data, the uric acid-to-creatinine ratio might be a better predictive indicator for preeclampsia than uric acid alone.

Validity of the findings

Please review the conclusion. Based on the data, uric acid does not appear to be sensitive but demonstrates good specificity (Optimal Threshold: 239.00 μmolL; Sensitivity: 0.39, Specificity: 0.85). This finding suggests that uric acid may be more effective as a specific rather than a sensitive risk factor in clinical decision-making.

Please review Table 4 "Distribution of renal function indicators among cases with or without adverse outcomes". The sample sizes for both groups are reported identical: n=23 for cases with and without adverse maternal outcomes, and n=45 for cases with and without neonatal adverse outcomes. This raises the question of whether the grouping and analysis are accurate, or whether there might be an error in reporting.

In Table 2, "Characteristics of maternal and neonatal birth outcomes," it appears that 120 cesarean sections occurred in the preeclampsia group based on the provided dataset, rather than the 122 reported. Please verify the data and update the table accordingly to ensure accuracy.

In Table 4, the mean BUN value for the group without adverse maternal outcomes based on the provided data is 2.75 instead of 2.57. Please recheck.

---

## Round 0.2 · Minor Revisions

Dear authors,

The study entitled “The relationship between renal function indicators and preeclampsia in the second trimester of pregnancy: a retrospective study” demonstrated interesting findings using an appropriate methodological approach. However, minor revisions must be clarified in the manuscript. Your article has great potential for publication on PeerJ, but the reviewers have requested additional changes to be made.

Reviewer 1 ·

Basic reporting

no comment

Experimental design

no comment

Validity of the findings

no comment

Additional comments

The authors have addressed the comments from previous round comprehensively, making significant improvements to the manuscript. They revised figures for clarity and provided detailed explanations of preprocessing and feature extraction methods, enhancing reproducibility. The clustering approach was clarified, linking subtypes to biological and clinical relevance. Additional insights into biological pathways connecting renal function markers to preeclampsia were included, along with analyses of composite biomarker models to assess predictive accuracy. Language and grammatical improvements were made throughout the manuscript, and limitations were acknowledged with plans for multicenter validation and expanded future analyses. These updates significantly enhance the manuscript’s clarity and scientific rigor, making it ready for publication.

Reviewer 2 ·

Basic reporting

no comment

Experimental design

The study used a matched case-control design, aiming to remove confounders. However, the criteria for selecting the matching variables and the rationale for including only a limited number of variables remain unclear. It would be helpful to justify whether incorporating additional potential confounders through propensity score matching is necessary. Additionally, consider including references addressing the adequacy of confounding variable selection or discussing this as a limitation of the study.

Validity of the findings

no comment

Additional comments

no comment

·

Basic reporting

The comments have been taken into account by the authors. There are several issues concerning language.

Consider to replace "creatine" with "creatinine" in "while the serum creatine levels were higher in control group"

Replace "have" with "has" in "However, a systematic review have also shown that some renal function indicators"

Replace "MedClac" with "MedCalc"

Please rewrite "We hypothesize that the second trimester pregnancy carries a greater sensitivity than first trimester."

Replace "including those from the National Institute for health and care excellence, the American college of obstetricians and gynecologists," with "including those from the National Institute for Health and Care Excellence, the American College of Obstetricians and Gynecologists,"

Replace "Sensitivicy" with "Sensitivity" as y-axis label in Figure 2.

In Figure 1 contains violin plots but also a bar plot. What is the reason a single bar plot is included? Why not use violin plots solely?

Experimental design

No comments

Validity of the findings

No comments

---

## Round 0.3 · accepted · Accept

Dear Author,

Congratulations! After your diligent work addressing the reviewers' comments, I am pleased to inform you that your manuscript has been accepted for publication in PeerJ. This version is more concise and formal, enhancing clarity and flow.

Reviewer 2 ·

Basic reporting

NA

Experimental design

NA

Validity of the findings

NA

Additional comments

The authors successfully addressed my concerns and the manuscript is ready to publish.

·

Basic reporting

no comment

Experimental design

no comment

Validity of the findings

no comment